# Stereoselective chemical *N*-glycoconjugation of amines via CO₂ incorporation

Zihan Peng[1,3], Qian Xiao[1,3], Yan Xia[1], Mingyu Xia[1], Jia Yu[1,2], Pengfei Fang [1], Yu Tang [1] ✉ & Biao Yu [1] ✉

Chemical *N*-glycoconjugation can provide a unique way to tailor the properties of the ubiquitous amines for further expending their diverse functions and applications. Nevertheless, effective methodology for glycoconjugation of amines remains largely underdeveloped. Inspired by a biotransformation pathway of amine-containing drugs in vivo, we have developed an effective protocol that enables one-step chemical *N*-glycoconjugation of amines in high stereoselectivity under mild conditions. This protocol involves conversion of the amine moiety into the corresponding carbamate anion under CO₂ atmosphere and a subsequent SN2 type reaction with glycosyl halides. This work provides an example of using CO₂ as the coupling unit in chemical glycoconjugation reactions. A case study on the resulting *N*-glycoconjugates of Crizotinib, an anticancer drug, demonstrates a quick cleavage of the glucosyl carbamate linkage, testifying that this *N*-glyconjugation method could serve as a general approach to procure novel prodrugs.

Heteroatom (*O*-, *N*-, *S*-) glycoconjugation is a biologically and synthetically significant process, which provides the opportunity to tailor the properties of molecules for diverse functions and applications[1–11]. Given the ubiquitous occurrence of the *N*-containing structures in natural products, pharmaceuticals, and agrochemicals, the *N*-glycoconjugation of amines holds particularly great promise for modulating the physical, chemical, and biological properties of these compounds (Fig. 1A), which have indeed found numerous applications in medicinal chemistry and chemical biology[12–17]. However, chemical *N*-glycoconjugation of amines remains largely underdeveloped; indirect methods have been applied, which usually relies on the addition of a linker between the *N* atom and the glycosyl moiety[12,13,15,18]. Such approaches require pre-functionization of amines/glycosylating reagents and always suffer from multi-step operations and therefore not applicable for late-stage glycoconjugation. The development of convenient methods for one-step stereoselective *N*-glycoconjugation of amines therefore remains to be a challenging yet unmet goal.

1-*O*-Glycosyl carbamates, which combine amine and sugar through a carbonyloxy(-COO-) linkage, usually exhibit enhanced water solubility (compared to parent amines)[19] due to the presence of sugar moiety, are highly stable in neutral aqueous conditions, and can be enzymatically cleaved to release the parent amines in vivo[20], making this structural motif particularly valuable for applications in medicinal chemistry and chemical biology. Indeed, a variety of 1-*O*-glycosyl carbamates have been reported as prodrugs[21–30], biosensing agents[31–35], and lectin ligands[36–38]; some examples are shown in Fig. 1B. The role of the sugar moiety in the pharmacological activities of *N*-glycomodified amine-containing drugs have been studied previously, and in several cases, it was demonstrated that *N*-glycomodification significantly enhanced drug targeting and specificity[23,24]. A representative example is shown in Fig. 1C, where the Hecht group confirmed that the disaccharide moiety of bleomycines (BLM) facilitated its uptake by cancer cells, thus BLM carbamate glycoside (**5**) exhibited significantly more potent cytotoxic activity than deglycoBLM (**6**)[30]. The influence of anomeric configuration on the enzymatic cleavage of the 1-*O*-glycosyl carbamates have been showcased in a pioneering work by the Waldmann group[20], where they found that both the α- and β-glycosidase actively cleaved the α- and β-anomers of glycosyl carbamate **7**, respectively, with high stereospecificity (Fig. 1D). Thus, for the medical application of glycosyl carbamate compounds, anomeric purity is an important issue.

[1]State Key Laboratory of Chemical Biology, Shanghai Institute of Organic Chemistry, University of Chinese Academy of Sciences, Chinese Academy of Sciences, Shanghai 200032, China. [2]Key Laboratory of Structure-Based Drugs Design and Discovery of Ministry of Education, Shenyang Pharmaceutical University, Shenyang 110016, China. [3]These authors contributed equally: Zihan Peng, Qian Xiao. ✉e-mail: tangyu@sioc.ac.cn; byu@sioc.ac.cn

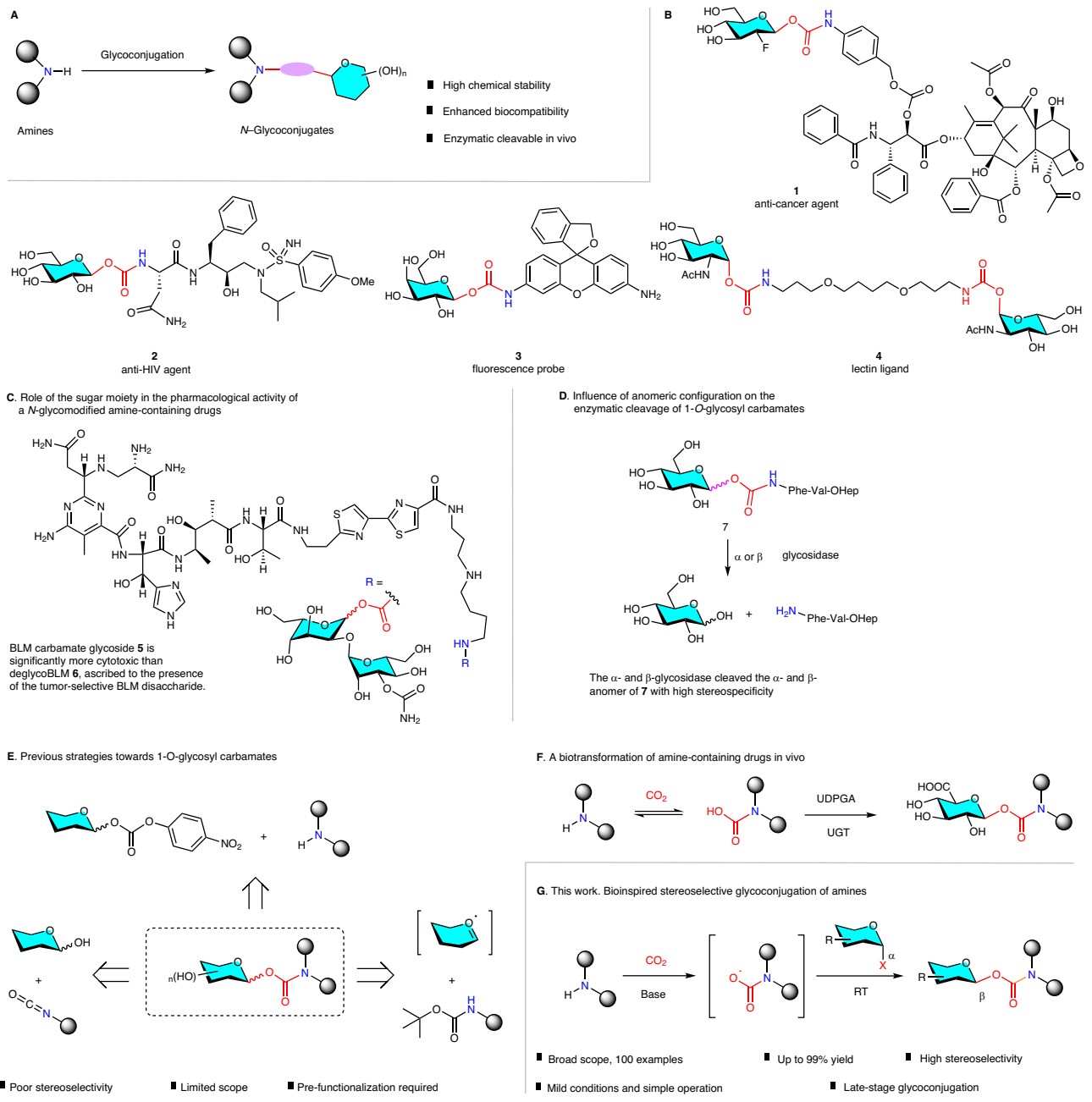

**Fig. 1 | An introduction to *N*-glycoconjugation of amines. A** General illustration. **B** 1-*O*-glycosyl carbamates, a unique form of glycoconjugated amines, and its representative applications. **C** The role of a sugar moiety in the pharmacological activity of a *N*-glycomodified amine-containing drug. **D** Influence of anomeric configuration on the enzymatic cleavage of 1-*O*-glycosyl carbamates. **E** Previous strategies towards 1-*O*-glycosyl carbamates. **F** An interesting biotransformation pathway of amine-containing drugs in vivo. **G** This work: bioinspired stereoselective glycoconjugation of amines. R, functional group. UDPGA, uridinediphosphate glucuronic acid, UGT, uridine diphosphate glucuronosyl transferase, RT, room temperature.

Several methods have been developed for the preparation of 1-*O*-glycosyl carbamates (Fig. 1E)[39–41]. Nevertheless, these methods suffer from several inherent drawbacks, including (1) poor anomeric stereoselectivity, (2) limited substrate scope, and (3) requirement of pre-functionalization steps. These drawbacks have seriously limited the real-world applications of 1-*O*-glycosyl carbamates. Thus, the development of new methodologies toward convenient and stereoselective synthesis of 1-*O*-glycosyl carbamates is highly desirable.

During metabolomics studies of amine-containing drugs, an intriguing biotransformation pathway has been identified, in that a primary or secondary amine-containing drug is capable of reacting with carbon dioxide ($CO_2$) via a reversible reaction to form a carbamic acid, which undergoes glucuronidation catalyzed by UDP-glucuronyl transferase (UGT) with uridinediphosphate glucuronic acid (UDPGA) as a source of the glucuronyl moiety, leading to the formation of a stable carbamate glucuronide metabolite (Fig. 1F)[42]. Inspired by this biotransformation, we envisioned a possible route to 1-*O*-glycosyl carbamates through a one-pot stereoselective coupling of amines, $CO_2$, and glycosylating reagents, which once achieved would constitute a conceptually simple and practically appealing method for stereoselective *N*-glycoconjugation of amines. Herein, we report the development of this *N*-glycoconjugation method (Fig. 1G), its scope, and applications to the chemical *N*-glycoconjugation of various amine-containing drugs.

## Results

Under basic conditions, amines can readily and reversibly react with $CO_2$, leading to the formation of carbamate anions[43–52]. We hypothesized that an electrophilic glycosylating reagent might be able to trap the resulting carbamate anion, leading to the formation of 1-*O*-glycosyl carbamates (Fig. 1G). Glycosyl halides were selected as the electrophilic glycosylating reagents due to their ease of access, high anomeric purity (usually occur as pure α-anomers), and high electrophilic reactivities[53–55]. To test this hypothesis, model reactions were set up using (*R*)−1-(1-naphthyl)ethylamine (**A1**) and *N*-benzoylpiperazine (**A2**) as model primary and secondary amine substrates and glucopyranosyl halides (**G1-G4**) bearing varied protecting groups and halide ions as model glycosylating reagents. Extensive optimization of reaction conditions revealed that when first reacting amines **A1** or **A2** (1.0 eq) with $CO_2$ (1 atm) in the presence of $Cs_2CO_3$ (1.5 eq) in DMSO (0.25 M) at RT for 2 h, followed by addition of perbenzyl glucosyl α-chloride **G4** (3 eq, for **A1**) or peracetyl glucosyl α-bromide **G2** (2 eq, for **A2**) and further reaction at RT for 12 h could provide the desired *N*-glycoconjugated product **GA2** (75%; Fig. 2, entry 4) and **GA12** (>99%, Fig. 2, entry 15) in satisfactory yields. It was observed that the anomeric configuration and the types of protecting groups considerably influenced the coupling efficiency. For primary amines, optimal results were obtained only when using perbenzyl glycosyl α-chloride (Fig. 2, entries 1-4), whereas for secondary amines, perbenzyl glycosyl α-chloride and peracetyl glycosyl α-bromide performed equally well (Fig. 2, entries 14-17). NMR analysis revealed that when using glycosyl α-halides, the *N*-glycoconjugated products were all pure β-anomers, suggesting $S_N2$ character of the reaction.

With these feasible conditions in hand, a series of glycosyl halides (**G5-G29**) were prepared and subjected to the reaction conditions to examine the glycosyl halides scope of this transformation, and the results were summarized in Figs. 2 and S29. We were pleased to observe that a variety of glycosyl chlorides, including galactopyranosyl- (**G7**), 2-deoxy-2-fluoro-glucopyranosyl- (**G8** and **G10**), xylopyranosyl- (**G17**), and arabinopyranosyl- (**G21**) chlorides were suitable substrates for conjugation of primary amines, delivering the expected glycosyl carbamates in synthetically useful yields (43%–51%; Fig. 2, entries 6-8, 10 and 12,). With respect to secondary amines, an even broader glycosyl halides scope was observed; capable donors included galactopyranosyl bromide and chloride (**G5** and **G7**), fluorinated glucopyranosyl bromides (**G9** and **G12**), methyl glucopyranosyluronate bromide (**G13**), xylopyranosyl chloride and bromide (**G17** and **G18**), fucopyranosyl and arabinopyranosyl chlorides (**G19** and **G21**), and disaccharide cellobiosyl bromide (**G15**). A common feature of these successful substrates was that they were all 1,2-*cis*-substituted pyranosyl chlorides/bromides with a 2-equatorial oriented *O*- or *F*-atoms (see Fig. S29). However, 1,2-*cis*-perbenzylated pyranosyl α-bromides, such as **G11** and **G14**, 1,2-*trans*-substituted mannosyl α-iodide (**G22**), 2-deoxy-2-*N*-substituted-pyranosyl α-halides, including 2-deoxy-2-AcNH-substituted α-chloride and bromide and 2-deoxy-2-azido-substituted α-bromide and iodide (**G23-G26**), glycosyl fluoride (**G27**), and furanosyl halide (**G28, G29**), were found not suitable for the present transformation (Fig. 2, entries 32-39 and Fig. S29). Poor results were observed when using α-glycosyl iodides, such as **G3, G6, G22** and **G26** and perbenzylated α-fucosyl chloride (**G20**), owing to rapid decomposition of these poorly stable glycosylating halides under the conjugation conditions. It was noted that when using α-D-xylopyranosyl chloride/bromide as the glycosylating reagents, the product contained a small amount of the undesired α-anomer (Fig. 2, entries 26 and 28), presumably caused by an in situ anomerization process of the glycosyl halides catalyzed by the leaving halides ions under the coupling conditions[56]. To circumvent this problem, $Ag_2CO_3$ (1.0 eq) was added as a halide ion scavenger, to our delight, the formation of the undesired α-anomer was completely inhibited, leading to the pure

β-product in 90% yield (Fig. 2, entry 27). Apart from glycosyl halide, glycosyl epoxide (**G30**), another important type of glycosyl electrophile, was also tested, and no desired *N*-glycoconjugated product was detected (Fig. 2, entry 40).

Having established optimal reaction conditions and glycosyl halide scope, we next performed a series of experiments to probe the reaction mechanism (Fig. 3). In order to probe the relative reactivity of different glycosyl halides toward a same carbamate anion and the relative reactivity of carbamate anions derived from either primary or secondary amine toward a same glycosyl halide, a panel of competition experiments were performed (Fig. 3A). With the aid of deuterium labelling methods, it was confirmed that glucopyranosyl bromide exhibited much higher reactivity than glucopyranosyl chloride bearing the same protecting groups. (Fig. 3A-a). It was also found that the disarmed peracetyl glucopyranosyl chloride **G1** exhibited higher reactivity than the armed perbenzyl glucosyl chloride **G4** (Fig. 3A-b). The fact that the electron-deficient disarmed glucosyl halide exhibited higher reactivity than the electron-rich armed glucosyl halide indicated that the reaction processed via a direction substitution mechanism ($S_N2$ process) rather than a dissociation mechanism ($S_N1$ process). The peracetyl glucosyl bromide **G2** exhibited higher reactivity than the galactosyl counterpart **G5** (Fig. 3A-c), whereas **G2** and 2-deoxy-2-fluoro-glucosyl bromide **G9** exhibited similar reactivity (Fig. 3A-d). The carbamate anion derived from secondary amine **A2** exhibited much higher reactivity than the carbamate anion derived from primary amine **A1** (Fig. 3A-e). The fact that for less reactive carbamate anion derived from primary amine, less reactive perbenzyl glucosyl chloride **G4** exhibited much better coupling efficiency than the higher reactive peracetyl glucosyl halides **G2** and **G3** (Fig. 2, entries 2-4) and that for higher reactive carbamate anion derived from secondary amine, glucosyl bromide **G2** exhibited better coupling efficiency than the higher reactive glycosyl iodide **G3** (Fig. 2, entries 16-17) indicated that a "match" scenario of reactivity was the key to achieve good coupling results[57]. The carbamate anion exhibits lower nucleophilicity and thus, it shows better reactivity with less active but more stable glycosyl halides. Since the reaction environment is alkaline, more reactive glycosyl halides, such as glycosyl iodides or fully benzyl-protected glycosyl bromides, decompose quickly in the reaction system, thus exhibits poor reaction efficiency. The reactivity of the glycosyl halides bearing equatorial anomeric C-X bond were next studied using **G31** as a model substrate, poor yield and an anomeric mixture of products were obtained (Fig. 3B). This result is consistent with those in the literature, where it was found that β-glucosyl chlorides exhibited much poor $S_N2$ reactivity than its α-counterpart toward carboxylate anion nucleophiles[57,58].

An in-depth mechanistic study was next performed using amine **A2** and glucosyl chloride **G1** as model substrates. NMR analysis revealed that under 1 atm $CO_2$ atmosphere, **A2** was completely converted into the stable carbamate anion intermediate after 2 h (Fig. 3C and Fig. S40). The α-deuterium kinetic isotope effects were determined[59] to reveal an α-DKIE value of 0.87, which is consistent with previously reported α-DKIE values observed for a typical $S_N2$ process (Fig. 3D)[60]. Initial rate kinetic analysis revealed that the reaction was first order for the glycosyl chloride **G1** and the amine **A2**, showing a typical kinetics of the $S_N2$ reaction (Fig. 3E). Taken together, these results supported that the reaction proceeded via a direct $S_N2$ substitution mechanism.

The optimized conjugation conditions were then applied to the synthesis of a range of *N*-glycoconjugated amines using various amines with glucosyl halides **G1, G2**, or **G4** as the glycosylating reagents (Figs. 4 and S30). We were glad to find that a wide variety of primary and secondary aliphatic amines reacted smoothly under the present reaction conditions to yield *N*-glycoconjugated products (**GA35–GA64**) with complete β-stereoselectivity and good yields (38% −99%). Primary amines with the amino group appended to various rings (e.g., aza-2-cycloheptanone, cyclooctane, cyclohexyl, and piperidine) or α,α-disubstituted primary amines or even α-tertiary

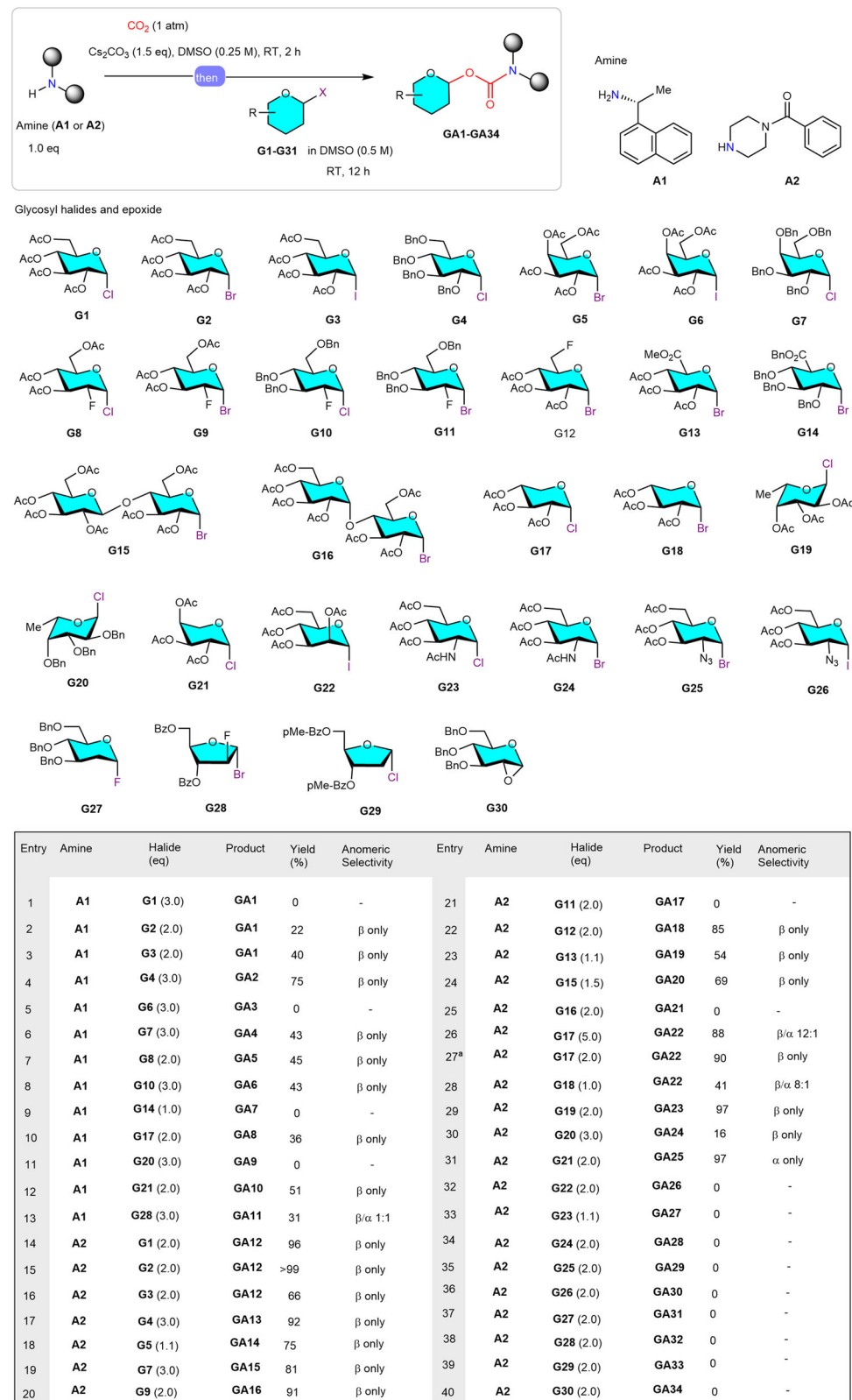

**Fig. 2 | Scope of glycosyl halides.** [a] in the presence of 1.0 eq of Ag$_2$CO$_3$. See the supporting information for experimental details, yields are the isolated yields, the α/β ratio was determined based on $^1$H NMR spectroscopic analysis of the purified mixture of anomeric stereoisomers. DMSO, dimethyl sulfoxide; RT, room temperature; R, functional group; Me, methyl; Bn, benzyl; Ac, acetyl; Bz, benzoyl; p-Me-Bz, 4-methyl-benzoyl.

primary amines all worked well. Both acyclic and cyclic secondary amines, including 5-membered, 6-membered, and bridged bicyclic amines worked similarly well. A series of common functional groups, such as methoxyl (**GA50** and **GA56**), vinyl (**GA51**), nitro (**GA57**),

cyano (**GA58**), and trifluoromethyl (**GA63**) groups, and pharmaceutically important motifs, including indole ring (**GA35** and **GA49**), pyridine (**GA60** and **GA62**), dibenzo[b,f][1,4]thiazepine (**GA61**), and 1,2,4-triazole (**GA63**) were well tolerated. To our delight, this

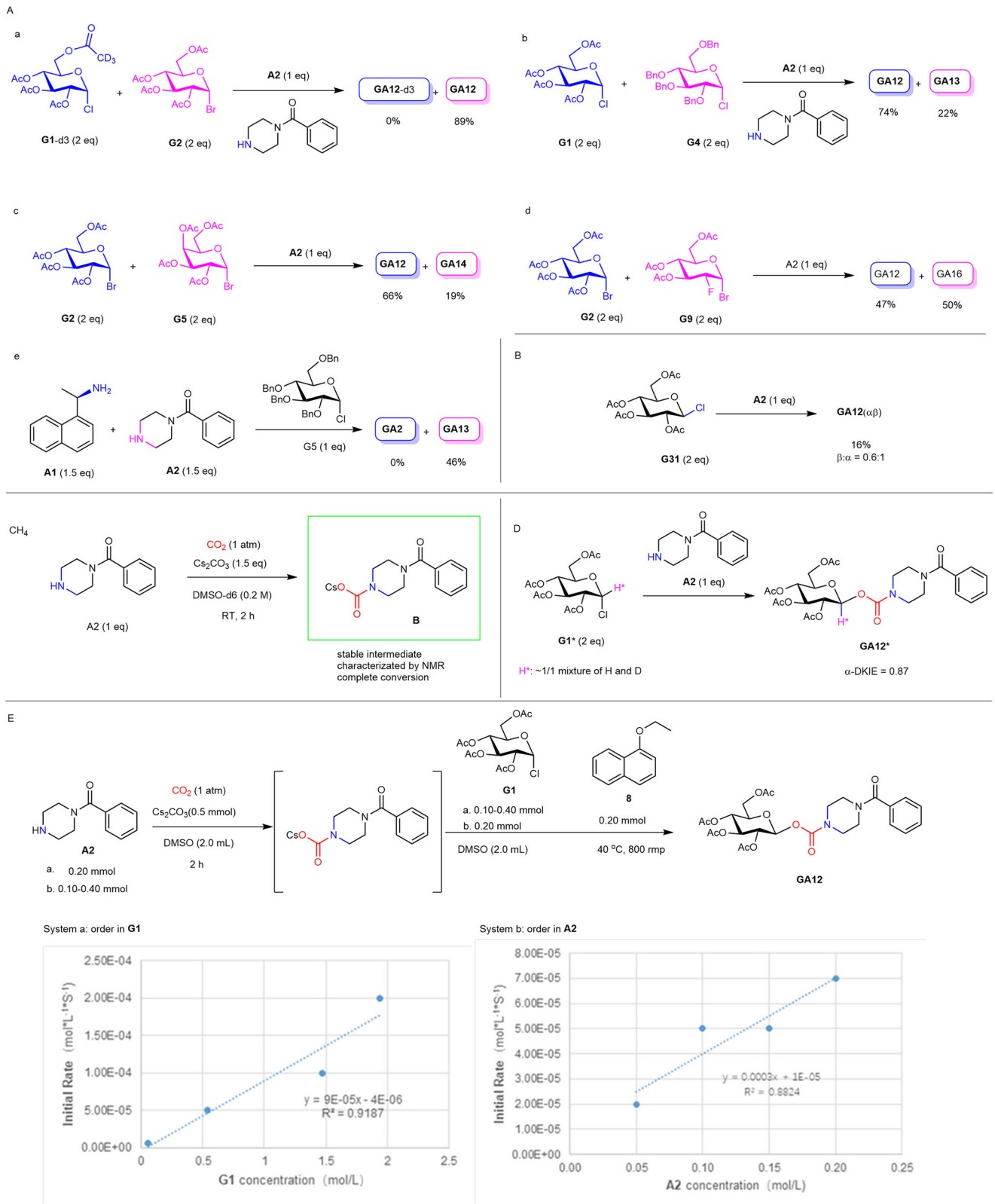

**Fig. 3 | Mechanistic studies. A** Competition experiments. **B** Reactivity of an equatorial anomer of glycosyl chloride. **C** NMR experiments. **D** Secondary α-deuterium kinetic isotope effects determination. **E** Kinetic studies. See the supporting information for experimental details. DMSO dimethyl sulfoxide, RT room temperature, Bn benzyl, Ac, acetyl.

method has also been proven effective to the chemical N-glycoconjugation of amino acids, leading to the access to a series of amino acid-sugar conjugates (i.e., **GA41-GA47, GA49**, and **GA54**) in synthetically useful yields (38%–86%). A dipeptide substrate, L-phenylalanyl-L-phenylalanine methyl ester, was smoothly N-glycoconjugated using this method (**GA48**), showcasing the potential of the present method in constructing peptide glycoconjugates. It should be mentioned that previous reports for the preparation of this type of amino

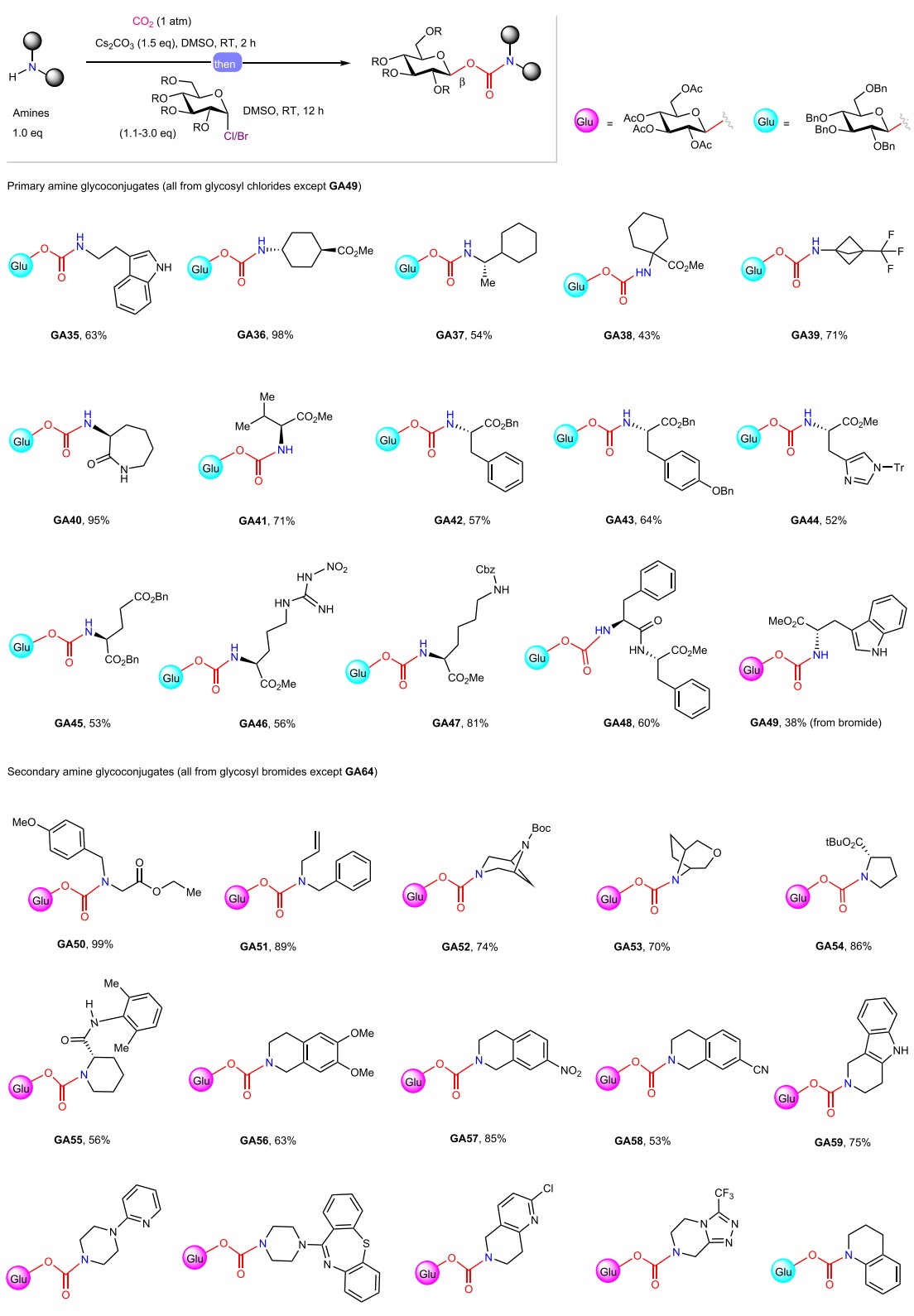

**Fig. 4 | Scope of amines.** See the supporting information for experimental details, yields are isolated yields, the α/β ratio was determined based on ¹H NMR spectroscopic analysis of the purified products. DMSO, dimethyl sulfoxide; RT, room temperature; R, functional group; Me, methyl; Bn, benzyl; Ac, acetyl; Tr, triphenylmethyl; Cbz, benzyloxycarbonyl; Boc, *tert*-butoxycarbonyl; tBu, *tert*-butyl.

acid-sugar conjugates all afforded a mixture of the α,β-anomers[19,20]. The present method represents the first example of stereoselective synthesis of this type of compounds. Primary aryl amines and secondary open chain *N*-alkyl anilines such as *N*-ethylaniline,

were found not suitable substrates (for unsuccessful examples, see Supplementary Fig. S6), whereas secondary cyclic *N*-alkyl anilines, such as 1,2,3,4-tetrahydroquinoline, were suitable substrates, leading to the expected *N*-glycoconjugated product **GA64** in 55% yield.

To further demonstrate the synthetic utility of this method, scale-up reactions were performed. Thus, compound **GA12** and **GA40** could be prepared at 10 mmol scale without loss of synthetic efficiency, delivering the expected product in 97% and 88% yields, respectively (Figs. S53 and S54). Moreover, the protecting groups of Ac-, Bn- and Boc- in product **GA12, GA40** and **GA102** could be selectively and efficiently cleaved using the conditions of $K_2CO_3$/MeOH, Pd-C/$H_2$ (1 atm) and TFA/$CH_2Cl_2$, respectively (Figs. S55 and S57).

The good functional group tolerance observed from the reaction scope studies encouraged us to explore the application of this *N*-glyco-conjugation protocol to late-stage functionalization of complex amine-containing bioactive molecules. To this end, a series of amine containing drugs and drug candidates, which possess diverse structural scaffolds with a broad spectrum of bioactivities, were subjected to the present reaction conditions. Gratifyingly, the desired *N*-glycoconjugation performed well in most of the tested cases, providing total 36 unprecedented *N*-glycoconjugated amine-containing drugs and drug candidates (Figs. 5 and S30). Notably, sitagliptin (**GA82**), vortioxetine (**GA69**), niraparib (**GA85**), crizotinib (**GA89**), and duloxetine (**GA72**), which are among the top 200 small molecule drugs by retail sales in 2022 (https://sites.arizona.edu/njardarson-lab/top200-posters/), could by effectively *N*-glycoconjugated with good efficiency (β only, 44−98%). These successful examples clearly demonstrated the application potential of this protocol in medicinal chemistry and chemical biology and will promote further exploitation of amine-sugar conjugates as potential therapeutic agents.

To showcase the potential value of the *N*-glycoconjugated amines prepared by the present method, two glucose-conjugates of crizotinib, a first-generation small molecule anaplastic lymphoma kinase (ALK) inhibitors for treatment of non-small-cell lung cancer[61], were prepared and subjected to bioactivity assays (Fig. 6). Enzymatic assays revealed that glucose-conjugates of crizotinib **GA89** and **GA95** exhibited median inhibitory concentration ($IC_{50}$) of 1.1 and 0.3 μM, respectively, for ALK activity, being slightly less active than the parent crizotinib ($IC_{50} = 0.1$ μM; Fig. 6). Cell-based assays revealed that the peracetylglucose-conjugate of crizotinib (**GA89**) inhibited the growth of HEK-293T, SK-MEL-28, and SNU-5 cells with $IC_{50}$ values of 5 μM, 5 μM, and 45 nM, respectively, while the $IC_{50}$ values of the glucose-conjugate of crizotinib (**GA95**) were 25 μM, 25 μM, and 22 nM, respectively, showing comparable antitumor activities with the parent crizotinib ($IC_{50} = 6$ μM, 6 μM, and 15 nM, respectively). Moreover, pharmacokinetics of crizotinib and its glucoside **GA95** were measured in healthy male C57 mice (Figs. 6 & S63 and Tab. S6). After oral application of glucoside **GA95**, the plasma concentration of crizotinib was quickly increased to the similar level as the crizotinib group, and then the pharmacokinetic parameters of **GA95** became similar to crizobinib. These results demonstrated that the glucose carbamate linkage could indeed be cleavable in vivo, thus the present *N*-glyconjugation could serve as an effective approach to the development of a general type of prodrugs.

## Discussion

Inspired by a biotransformation pathway of amine-containing drugs in vivo, we herein have developed an efficient chemical *N*-glycoconjugation of amines in high stereoselectivity under mild conditions. The reaction proceeds via an $S_N2$ process between amine-derived carbamate anions and glycosyl halides, and the $S_N2$ mechanism were supported by a series of mechanistic experiments. The identification of a suitable glycosyl halide as glycosyl electrophiles, which exhibit a matched reactivity toward the amine derived carbamate anion nucleophiles is the key to achieve the successful $S_N2$ displacement. The mild reaction conditions and high chemo- and stereoselectivity have permitted the *N*-glycoconjugation of 100 amine-containing compounds, including a series of complex pharmaceuticals. We expect that this protocol will find immediate applications in medicinal chemistry and chemical biology relevant to carbohydrates and foster the discovery of novel glycoconjugate therapeutics.

## Methods

### General procedure for the stereoselective *N*-glycoconjugation reaction

An oven-dried Schlenk tube (25 mL) were charged with amine (0.50 mmol, 1.0 eq), $Cs_2CO_3$ (0.75 mmol, 1.5 eq), and a Teflon-coated magnetic stirring bar. After the mixture was evacuated and backfilled with $CO_2$ gas three times, DMSO (2.0 mL, 0.25 M) was added via a syringe. The reaction mixture was allowed to stir for 2 h under the $CO_2$ atmosphere at RT, and then a solution of glycosyl halide (1.00 mmol, 2.0 eq) in DMSO (2 mL, 0.50 M) was added. The mixture was further stirred at RT for 12 h, and then $H_2O$ (10 mL) was added. The mixture was extracted with $CH_2Cl_2$ (20 mL × 3), washed with saturated $NH_4Cl$ (50 mL), then dried over anhydrous $Na_2SO_4$ and filtered. The solvent was removed in vacuo, and the residue was purified by silica gel column chromatography to afford the desired product.

### Note

Occasionally, gelation occurred when using $Cs_2CO_3$ as the base, leading to diminished yields. To circumvent this problem, $K_2CO_3$ could be used instead.

### Cell viability assay

Cancer cell lines were procured from the National Collection of Authenticated Cell Cultures, China. All cells were kindly provided by the Cell Bank of the Chinese Academy of Sciences and have been certified through STR analysis. All cells were tested using HiScript III All-in-one RT SuperMix Perfect for qPCR (Vazyme, Cat: R333-01), and the test results were negative for mycoplasma contamination. These cells were cultured in Dulbecco's Modified Eagle's Medium (Gibco), supplemented with 100 U/mL penicillin and 100 U/mL streptomycin (Gibco). The cells were hemi-depleted each week with fresh medium and maintained at $3 \times 10^5$ cells/mL at 37 °C and 5% $CO_2$. Cell viability was analyzed by Cell Counting Kit-8 (CCK8, Beyotime, Shanghai, China) according to the manufacturer's protocols. Cells were seeded and cultured at a density of $3 \times 10^3$ /well in 100 μL of medium into 96-well microplates (Corning, USA). Then, the cells were treated with various concentrations of compounds (0, 5 nM, 30 μM, 50 nM, 300 nM, 500 nM, 3 μM, 5 μM, 30 μM, 50 μM or 0, 0.13 nM, 0.53 nM, 1.64 nM, 4.92 nM, 14.80 nM, 44.40 nM, 133 nM, 400 nM, 1200 nM). After treatment for 72 h, 10 μL of CCK-8 reagent was added to each well and then cultured for 4 h. All experiments were performed in pentaplicate. The absorbance was measured at 450 nm by a microplate reader (PerkinElmer, USA) using wells without cells as blanks. The proliferation of cells was expressed by the absorbance.

### Enzymatic inhibition assay

**Protein purification**[62]. The plasmid pCDNA3.1_8×His-tev-ALK (kinase domain, 1068−1410) was constructed and extracted from E. coli DH5α (DE3) strain by alkaline solution method. The plasmid was transiently transfected into 3 L Expi293 cells at a density of $3 \times 10^6$ cells/mL using transfection reagent PEI (25,000 Da). The cells were collected by centrifugation on the 5th day after transfection, and lysed at 600 bar at 4 °C. The centrifugation supernatant of cell lysate was loaded to a pre-balanced Ni-NTA column and washed with 100 mL solution containing 25 mM Tris pH 8.0, 150 mM NaCl, and 25 mM imidazole. The ALK protein was then eluted with 40 mL solution containing 25 mM Tris pH 8.0, 150 mM NaCl, and 250 mM imidazole. The eluent was concentrated to 500 μL and further purified by a Superdex 200 column with the running buffer containing 25 mM HEPES pH 7.4 and 250 mM NaCl. The peak fractions were used for enzymatic assays.

**ATP hydrolysis assays**. The ATP hydrolysis assays were based on Kinase-Glo luminescent Kit (Promega). The experiment was carried out in

Corning 384-well white flat bottom microplates. The buffer in the reaction contains 25 mM HEPES pH 7.4, 140 mM NaCl, 40 mM $MgCl_2$, 30 mM KCl, 1 mM DTT, 0.1 mg·mL$^{-1}$ BSA, and 0.004% Tween-20. The compounds were diluted to different concentrations with a 5-fold gradient.

2.5 µL 100 nM ALK in HEPES buffer was mixed with crizotinib, **GA89**, or **GA95**, and incubated for 10 min. Then 5 µL substrate (2 µM ATP) was added to start the reaction at 37 °C and incubated for 6 h. DMSO was added into negative control wells. When the reaction

**Fig. 5 | Late-stage *N*-glycoconjugation of amine-containing drugs.** See the supporting information for experimental details, yields are isolated yields, the α/β ratio was determined based on $^1$H NMR spectroscopic analysis of the purified products.

DMSO, dimethyl sulfoxide; RT, room temperature; Me, methyl; Bn, benzyl; Ac, acetyl; *t*Bu, *tert*-butyl.

Crizotinib glucosides
**GA89** (R = Ac); **GA95** (R = OH)

| Compound | Kinase ALK IC$_{50}$ (µM) | HEK-293T IC$_{50}$ (µM) | SK-MEL-28 IC$_{50}$ (µM) | SNU-5 IC$_{50}$ (nM) | t$_{1/2}$ (h) |
|---|---|---|---|---|---|
| GA89 | 1.1 | 5 | 5 | 45 | N.D. |
| GA95 | 0.3 | 25 | 25 | 22 | 3.18[a] (0.726[b]) |
| Crizotinib | 0.1 | 6 | 6 | 15 | 3.02 |

**Fig. 6 | Bioactivities and pharmacokinetic parameter of crizotinib glucosides.** Pharmacokinetic parameters were tested after oral application of crizotinib (50 mg/kg) and **GA95** (91.6 mg/kg) to healthy male C57 mice, results are the mean ± SD of 3 subjects in each group. [a] Analyte: crizotinib; [b] analyte: **GA95**. Me, methyl; Ac, acetyl.

reaches the planned time, 10 µL diluted Kinase-Glo reagent (1/50 diluted with a buffer containing 50 mM Tris pH 7.5 and 5% glycerol) was added and incubated for 15 min. Luminescence was measured on the Magellan plate reader (Tecan) and nonlinear regression was performed with Prism (GraphPad).

**Pharmacokinetic analysis of crizotinib glucosides.** All procedures in the animal studies were performed in accordance with the Guide for the Care and Use of Laboratory Animals of Shanghai Institute of Materia Medica, Chinese Academy of Sciences.

The animals were maintained in cages at 22 ± 3 °C and 55% relative humidity under a 12 h dark/light cycle. Rats were allowed free access to water but were fasted for 12 h before drug administration. The following protocol is typical for evaluating the pharmacokinetic characteristics of the test molecules in male C57 mices. The animals were deprived from food over a time period of 12 h prior to administration and 4 h after administration of the test molecules. Water was supplied without limitation. On the study day, the animals received test molecule (**G89, GA95** or crizotinib) by oral gavage, formulated in mixtures of 0.5%CMC-Na. Then blood was drawn from the retro-orbital venous following time points: 0.25, 0.5, 1, 2, 4, 6, 8 and 24 h after dosing. Plasma was obtained by centrifugation at 11,000 g for 10 min and stored at −80 °C until analysis.

Circulating concentrations of test compounds were determined using LC/MS/MS methods. The analytical range in mouse plasma was linear over a concentration range of 1.00-8000 ng/mL for crizotinib, 1.00-8000 ng/mL for **GA89**, 1.00-8000 ng/mL for **GA95**. Pharmacokinetic parameters were calculated from concentration versus time data using noncompartmental pharmacokinetic methods using Phoenix pharmacokinetic software.

**Reporting summary**
Further information on research design is available in the Nature Portfolio Reporting Summary linked to this article.

## Data availability
The data reported in this paper are available within the article and its Supplementary Information files. All data are available from the corresponding author upon request.

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

## Acknowledgements

Financial support from the National Natural Science Foundation of China (Grant NO. 22371291 (Y.T.) and 22031011 (B.Y.)), National Key R&D Program of China (Grant NO. 2022YFA1304700 (B.Y.)), Key Research Program of Frontier Sciences of the Chinese Academy of Sciences (Grant NO. ZDBS-LY-SLH030 (B.Y.)), the Strategic Priority Research Program of the Chinese Academy of Sciences (Grant NO. XDB1060000 (B.Y.)), Youth Innovation Promotion Association of the Chinese Academy of Sciences (Grant NO. 2021251 (Y.T.)), CAS Project for Young Scientists in Basic Research (Grant NO. YSBR-095 (Y.T.)), and Shanghai Municipal Science and Technology Major Project (B.Y.) is acknowledged.

## Author contributions

Y.T. and B.Y. designed the research and experiments. Z.P., Q.X., and J.Y., performed the synthetic experiments. Y.X., M.X. and P.F. performed the bioassays. Y.T and B.Y. wrote the paper. B.Y. conceived and supervised the project. All authors discussed the results and commented on the paper.

## Competing interests

A provisional patent application (application number: 2024105976875) naming Y.T., B.Y., Z.P., Q.X., and Y.X. as inventors has been filed by the Shanghai Institute of Organic Chemistry, CAS, which covers the synthetic method and medical applications described in this manuscript. The remaining authors declare no competing interests.
