## [Transparent Peer Review file · Nature Communications]

Stereoselective Chemical N-Glycoconjugation of Amines via CO₂ Incorporation

Corresponding Author: Professor Biao Yu

Version 0:

Reviewer comments:

Reviewer #1

(Remarks to the Author)

In the current manuscript, Yu, Tang, and co-workers disclosed a one-step chemical N-glycoconjugation strategy for primary and secondary amines. Key steps involve the in situ generation of carbamate anion from free amines and CO₂ and followed SN₂ type nucleophilic substitution with glycosyl halides. The exceptional mild reaction conditions, broad substrate scope, excellent functional group tolerance, simple operation, moderate to high yields, high stereoselectivity, and substrate accessibility make this method exhibit significant practical application potential. The practical synthetic potential was demonstrated by the late-stage functionalization of amine-containing drugs or drug candidates. The bioactivity assays and pharmacokinetic studies of the glucosides of two drug molecules proved the current N-glycoconjugation an effective approach to developing prodrugs.

Overall, the current protocol provides a facile method for accessing 1-o-glycosyl carbamates, which may serve as a practical general type of prodrugs. This referee support its publication on Nat. Commun. after the following comments being addressed.

- 1) The reaction between carbamate anions and glycosyl halides should be substitution reaction. The description of "addition" in the abstract is misleading.
- 2) To further demonstrate the practical application potential, large-scale synthetic experiments should be conducted.
- 3) The production of the small amount of stereoisomers (Figure 2, entries 26-27) was attributed to the leaving halide ions induced in situ anomerization process. What would happen if silver salt was added as an additive to trap the leaving halide ions?
- 4) It's well known that Lewis acids and Bronster acids could effectively promote the substitution reaction, will it work for the failed examples in the current manuscript? For example entries 6, 10, 12, and 22-26.
- 5) If the Lewis acids or Bronster acids can promote the reaction, would chiral ligands or chiral Bronster acids induce or change the stereoselectivity?
- 6) In lines 140 to 141, the explanation of "a match" scenario of reactivity" was ambiguous, deeper and more detailed reasons should be provided.

Reviewer #2

(Remarks to the Author)

The manuscript "Stereoselective chemical N-glycoconjugation of amines via CO₂ incorporation" describes the development of an effective methodology for glycoconjugation of amines, which is particular significance to modern chemical synthesis and pharmaceutical chemistry. The manuscript provides interesting data on bioinspired stereoselective glycoconjugation of amines, yet it is not ready for publication in nature communications. The manuscript might be accepted after major revisions. If you modify the manuscript point by point.

1. It is recommended that the authors summarize and condense the effect of the types of glycosyl donors and protecting groups on the reaction yield and stereoselectivity.

2. Based on the authors' attempts and findings on the screening and reaction of glycosyl donors and various types of amino acceptors, although the glycoconjugation of amines is affected by many factors, it is strongly recommended that the authors carry out mechanistic investigations on the basis of the results of a large number of experiments and draw a diagram of the

reaction mechanism process, and the optimal case can be selected for the mechanistic validation experiments and analysis.

Reviewer #3

(Remarks to the Author)

Comments and Suggestions

The manuscript by Yu and coworkers presents the development of an effective glycoconjugation method in which sugar and amine are coupled through a carbamate linkage. This is the first instance of using CO₂ gas in carbohydrates to transform the anomeric position of sugar. The authors have explored a variety of glycosyl halides and demonstrated that the stereochemistry of the carbamate linkage depends on the types and stereochemistry of the anomeric halides. Additionally, they have shown the efficiency of this new methodology by conducting a large number of successful transformations. However, I have a few major and minor concerns which are as follows:

Major comments:

1. The authors have emphasized that conjugating amines to sugar allows for the exploration of the diverse properties and functions of amines. Sugar chemistry is complex and involves multiple steps to prepare suitable glycosyl halides, especially oligosaccharides. I do not see any merit in including sugar as a trapping electrophile for carbamate anions. Multiple chemical methods are known to protect amines as carbamates. This point raises a valid concern as to how the presented study would find practical applicability in carbohydrate chemistry.
2. The authors have claimed to have achieved high stereoselectivity of the anomeric carbamate linkage. I would like to know the significance of the anomeric purity of the carbamate linkage in sugar chemistry or chemical biology. It would be interesting to cite relevant papers that discuss the significance of α - or β -anomers having carbamate linkage.
3. The authors have proposed carbamate linkage could offer a potential route to synthesize prodrugs. It would be interesting to discuss the kinetics of enzyme-catalyzed hydrolysis of α - and β -carbamate linkages, given that the authors have primarily constructed β -linkages.
4. One section of this manuscript has been dedicated to constructing glycoconjugates using amines as drug molecules. The authors should show at least few examples where sugar confers cell-specific targeting, controlled release or improve cellular internalization.
5. Have the authors tried trapping the carbamate ions with other electrophiles such as alkyl halides (Cl, Br, I), carbocations, carbonyl compounds, sulfonium ions, or nitronium ions?

Minor Comments:

1. In lines 59 and 60, the authors mentioned that the carbamate linkage enhanced the water solubility of molecules. Please provide specific references that discuss similar molecules where water solubility has been enhanced after the introduction of a carbamate linkage.
2. In lines 90 and 91, "halide ions" either can be removed since glucopyranosyl halides is generic terms for all halides or be specific which halides has been used.
3. In lines 98-100, the authors describes "It was observed that the anomeric configuration, types of protecting groups, and halide ions in the glucosyl donors considerably influenced the coupling efficiency", the authors need to discuss the effects of different types of protecting groups and their positions on sugar molecules, particularly how these factors influence the stereochemistry and yield of the products.
4. The authors have drawn a conclusion of SN₂ character of the reaction, most of the halides used in the manuscript are α -anomer. It is generally accepted that β -anomer (halides) is more reactive than α , it would be fascinating to know if β -anomeric halides are used SN₂ character of the reaction would be retained. There are multiple methods known for preparing β -anomeric halides, such as the bromination of thioglycosides.
5. The conclusion section is poorly written, with significant portions being repeated verbatim from the abstract, such as the statement, "The mild reaction conditions and high chemoselectivity have allowed successful N-glycoconjugation of 100 amines, including 36 drug molecules."

Including all the comments and suggestions would have a broad impact considering the general readership of Nature Communications journal. The potential publication of this manuscript in Nature Communications is only recommended if all the comments and suggestions are addressed.

Reviewer #4

(Remarks to the Author)

In this manuscript, the authors present a new method for N-glycoconjugation of amine-containing drugs via a carbamate linkage. The process involves pre-treating the amines under a CO₂ atmosphere in the presence of bases, followed by reaction with glycosyl halides in the same flask at room temperature. Demonstrated with over 100 examples, the reaction is highly anomerically selective and straightforward, representing a significant improvement over existing methods for preparing similar carbamate-linked N-glycoconjugates.

Additionally, the authors show that the corresponding glucose-conjugate of Crizotinib can be hydrolyzed in vivo to yield the parent Crizotinib, with comparable antitumor activity to the parent drug. Given that some sugars are used as directing groups for drug delivery, this reviewer is curious whether a carefully selected sugar-conjugate could also influence the distribution of the conjugated drug. Moreover, could 1,2-anhydro sugars be used instead of glycosyl halides in this transformation?

Overall, this biotransformation-inspired glycoconjugation is highly anomerically selective, the reaction operation is not technically demanding, and the resulting glycoconjugates are enzymatically cleavable, potentially useful in medicinal chemistry. The manuscript is recommended for publication in Nature Communications after addressing the following questions.

1. Since the use of bases in the system, is there any analysis regarding the potential racemization of optically active compounds, such as α -amino acids or similar moieties in drugs?
2. Do the authors have any speculation about which type of enzyme catalyzes the hydrolysis of the N-glycoconjugates?
3. Line 59, it seems not "a carboxyl (-COO-) linkage".
4. As declared in line 115-117, "1,2-trans-substituted pyranosyl halides" and several other halides were not suitable for the titled transformation. Since G22 is bearing anomeric iodide, G25 is bearing anomeric bromide, corresponding anomeric chlorides have not been tested, it is better not to draw a general conclusion for 1,2-trans-substituted pyranosyl halides or 2-N-substituted pyranosyl halides.
5. Line 190, "... crizotinib was quickly at the similar level..." is not a complete sentence.
6. The organization of Table S1 needs to be reordered to better present the optimization strategy.
7. Some ^1H NMR spectra may need to be further purified such as GA1, GA2, GA4, GA122; and some NMR spectra have uneven baselines, such as ^{13}C NMR of GA4, GA13, GA65, GA69, GA73, GA77, GA80, GA81, 82, GA83, GA103.

Version 1:

Reviewer comments:

Reviewer #1

(Remarks to the Author)

The authors have addressed my concerns adequately. I support its publication now.

Reviewer #2

(Remarks to the Author)

The authors have responded well to the questions I asked. This work provides a new general approach for Chemical N-glycoconjugation modification of drugs. The experimental methodology is novel and rational, and the experimental results support the conclusions.

Reviewer #3

(Remarks to the Author)

The revised manuscript by Professor Yu and colleagues addresses most of the comments and suggestions provided. The authors have underscored the importance of the carbamate linkage by referencing research from the Hecht group, which demonstrates how drug uptake can be enhanced by sugar moieties.

However, I disagree with their rebuttal on the transformation of oligosaccharides into their peracetylated forms for the generation of anomeric halides. In my view, this transformation may require multiple steps depending on the protecting and anomeric groups present on the sugar, and thus, their justification for synthesizing anomeric halides from peracetylated oligosaccharides in a single step seems overly simplistic. Additionally, the deprotection of acetates might compromise the carbamate linkage.

Regarding the enhancement of drug solubility, the increase in solubility is not due to the carbamate linkage but rather the hydroxyl groups on the sugar moiety, which is an inherent characteristic of sugars (as also mentioned in reference 19). The way the authors have presented this in their paper is somewhat confusing, as it suggests that the improved solubility is a direct result of the carbamate linkage. However, I do not believe this issue warrants further explanation.

I have no additional comments or suggestions, and I recommend accepting the article without further revision.

Response to the comments and suggestions

Reviewer #1:

In the current manuscript, Yu, Tang, and co-workers disclosed a one-step chemical N-glycoconjugation strategy for primary and secondary amines. Key steps involve the in situ generation of carbamate anion from free amines and CO₂ and followed SN₂ type nucleophilic substitution with glycosyl halides. The exceptional mild reaction conditions, broad substrate scope, excellent functional group tolerance, simple operation, moderate to high yields, high stereoselectivity, and substrate accessibility make this method exhibit significant practical application potential. The practical synthetic potential was demonstrated by the late-stage functionalization of amine-containing drugs or drug candidates. The bioactivity assays and pharmacokinetic studies of the glucosides of two drug molecules proved the current N-glycoconjugation an effective approach to developing prodrugs.

Overall, the current protocol provides a facile method for accessing 1-o-glycosyl carbamates, which may serve as a practical general type of prodrugs. This referee support its publication on Nat. Commun. after the following comments being addressed.

We are deeply grateful for the comments provided by the reviewer regarding our manuscript. In the following section, we provide a point-by-point response to each of the comments. We hope that our responses adequately address your concerns.

1) The reaction between carbamate anions and glycosyl halides should be substitution reaction. The description of “addition” in the abstract is misleading.

Answer: This has been corrected.

2) To further demonstrate the practical application potential, large-scale synthetic experiments should be conducted.

Answer: Thanks for the suggestion. Two examples of scaled up reactions (at 10 mmol scale) have been added; please find in page 8, paragraph 2 and Supporting Information,

section 14 (Page 115).

3) The production of the small amount of stereoisomers (Figure 2, entries 26-27) was attributed to the leaving halide ions induced in situ anomerization process. What would happen if silver salt was added as an additive to trap the leaving halide ions?

Answer: We appreciate for raising this question. According to your suggestion, we added 1.0 equiv. of Ag_2CO_3 into a reaction system as halide ion scavenger; to our delight, the formation of the undesired α -anomer was completely inhibited, leading to pure β -product in 90% yield. This set of data has been added to the revised manuscript (Fig 2, entry 27). For details, please see page 6, paragraphs 1.

4) It's well known that Lewis acids and Bronster acids could effectively promote the substitution reaction, will it work for the failed examples in the current manuscript? For example entries 6, 10, 12, and 22-26.

Answer: Thanks for the suggestion. In this manuscript, we use glycosyl halides as the glycosylating reagent, and it is reported that certain types of Lewis acids such as FeCl_3 could indeed promote the *O*-glycosylation of glycosyl halides (*Org. Biomol. Chem.*, **2018**, *16*, 9133-9137). However, in our reaction system, using Lewis acids or Bronster acids as promoters is not feasible due to the following reasons: 1) the incompatibility of the acidic promoter and the bases such as Cs_2CO_3 present in the reaction system; 2) carbamate anions are stable only under basic conditions. Under acidic or neutral conditions, they rapidly decompose to the starting materials (amine and CO_2).

5) If the Lewis acids or Bronster acids can promote the reaction, would chiral ligands or chiral Bronster acids induce or change the stereoselectivity?

Answer: In recent years, significant advances have been made using chiral ligands or chiral Bronster acid catalyst to control the anomeric stereoselectivities of chemical glycosylation reactions (*Chem. Soc. Rev.*, **2018**, *47*, 681-701; *Science*, **2017**, *355*, 162-166; *Nature* **2022**, *608*, 74-79.). Thus, it is possible to use this strategy to control the anomeric stereoselectivities of the glycosylation reaction between carbamate anions

and a proper glycosyl donor (maybe glycosyl phosphate as donor), and this will be a subject of our further research.

6) In lines 140 to 141, the explanation of “a match” scenario of reactivity” was ambiguous, deeper and more detailed reasons should be provided.

Answer: Thanks for this suggestion. A summary schematic diagram of the scope of glycosyl halides has been provided (Supporting information, section 4, page 14), and a detailed discussion on this point has been provided. The carbamate anion exhibits lower nucleophilicity and thus, it shows better reactivity with less active but more stable glycosyl halides. Since the reaction environment is alkaline, more reactive glycosyl halides, such as glycosyl iodides or fully benzyl-protected glycosyl bromides, decompose quickly in the reaction system, thus exhibits poor reaction efficiency. For details, please see page 5, paragraph 2.

In addition, a series of control experiments have been conducted, and the results further supported the S_N2 mechanism. For details, please see page 6, paragraphs 2 and Fig. 3.

Reviewer #2:

The manuscript "Stereoselective chemical N-glycoconjugation of amines via CO₂ incorporation" describes the development of an effective methodology for glycoconjugation of amines, which is particular significance to modern chemical synthesis and pharmaceutical chemistry. The manuscript provides interesting data on bioinspired stereoselective glycoconjugation of amines, yet it is not ready for publication in nature communications. The manuscript might be accepted after major revisions. If you modify the manuscript point by point.

We are grateful for the comments provided by the reviewer regarding our manuscript. In the following section, we provide a point-by-point response to each of the comments. We hope that our responses adequately address your concerns.

1. It is recommended that the authors summarize and condense the effect of the types of glycosyl donors and protecting groups on the reaction yield and stereoselectivity.

Answer: Thanks for this suggestion. A summary schematic diagram of the scope of glycosyl halides has been provided (Supporting information, section 4, page 14), and a detailed discussion on the effect of the types of glycosyl donors and protecting groups on the reaction yields and stereoselectivities has been provided. For details, please see page 5, paragraphs 2, and revised Fig. 3.

2. Based on the authors' attempts and findings on the screening and reaction of glycosyl donors and various types of amino acceptors, although the glycoconjugation of amines is affected by many factors, it is strongly recommended that the authors carry out mechanistic investigations on the basis of the results of a large number of experiments and draw a diagram of the reaction mechanism process, and the optimal case can be selected for the mechanistic validation experiments and analysis.

Answer: We greatly appreciate this suggestion. Accordingly, an in-depth mechanistic study has been performed using amine **A2** and glucosyl chloride **G1** as model substrates, and the results are depicted in Fig. 4. Through a combination of NMR analysis, α -DKIE determination, and initial rate kinetic analysis, we could conclude that the reaction indeed proceeds via a direct S_N2 substitution mechanism. For details, please see page 7, paragraphs 2, and revised Fig. 3.

Reviewer #3:

The manuscript by Yu and coworkers presents the development of an effective glycoconjugation method in which sugar and amine are coupled through a carbamate linkage. This is the first instance of using CO₂ gas in carbohydrates to transform the anomeric position of sugar. The authors have explored a variety of glycosyl halides and demonstrated that the stereochemistry of the carbamate linkage depends on the types and stereochemistry of the anomeric halides. Additionally, they have shown the efficiency of this new methodology by conducting a large number of successful transformations. However, I have a few major and minor concerns which are as follows: We would like to extend our sincere appreciation for the thorough review of our manuscript and the insightful questions and suggestions that are raised. In the following

section, we provide a point-by-point response to each of the comments. We hope that our responses adequately address your concerns.

Major comments:

1. The authors have emphasized that conjugating amines to sugar allows for the exploration of the diverse properties and functions of amines. Sugar chemistry is complex and involves multiple steps to prepare suitable glycosyl halides, especially oligosaccharides. I do not see any merit in including sugar as a trapping electrophile for carbamate anions. Multiple chemical methods are known to protect amines as carbamates. This point raises a valid concern as to how the presented study would find practical applicability in carbohydrate chemistry.

Answer: We deeply appreciate for raising this important question. At the outset of this research, we envisaged scenarios for the application of the method in the field of chemical glycobiology and medicinal chemistry relevant to carbohydrates; indeed our present results could find broad application, given the following reasons:

1) Heteroatom (*O*-, *S*-, *N*-) glyco-modification is a biologically and synthetically very important process and a central research issue in carbohydrate chemistry, new methods of chemical heteroatom glycol-modification have been continuously developed during the past decades. Of the three major types of heteroatoms, *O*- and *S*- glyco-modification is relatively straightforward, usually by directly attaching the sugar moiety (usually at the anomeric position) to the heteroatom as the corresponding *O*- and *S*- glycosides. *N*-glycosyl modification, however, is complex and challenging. Two important types of *N*-containing structures including amides and *N*-heterocycles (such as nucleobases) can form stable *N*-glycosides and therefore can be glyco-modified through direct *N*-glycosylation. Amines, which represent another important type of *N*-containing structure, is not suitable for direct *N*-glycosylation modification due to the instability nature of the corresponding *N*-glycosides. Thus, for a long time in the field of carbohydrate chemistry, there lacks a convenient method for amine glyco-modification, and our present work addressed this long-standing problem.

- 2) Most glycosyl halides, including those derived from oligosaccharides, are in fact convenient to prepare from simple commercially available sugar build blocks such as peracetylated sugars in one step. Many of them are stable enough for handling and storage at room temperature. Some monosaccharide derived glycosyl halides are now commercially available at a relatively low price. Thus, the synthetic experiments depicted in our current work using glycosyl halides as glycosylating reagents is convenient to perform.
- 3) In recent years, carbamate glycosides have been applied in the field of medicinal chemistry and chemical biology. The anomeric configuration of the carbamate glycosides could significantly influence its enzymatic cleavage process, whereas the known synthetic methods are not generally stereoselective. Therefore, a new method for the stereoselective preparation of carbamate glycosides is highly desirable, and our present approach fulfils this need.

2. The authors have claimed to have achieved high stereoselectivity of the anomeric carbamate linkage. I would like to know the significance of the anomeric purity of the carbamate linkage in sugar chemistry or chemical biology. It would be interesting to cite relevant papers that discuss the significance of α - or β -anomers having carbamate linkage.

Answer: This and the following comments are very helpful for improving the structuring of the introduction of our work. In a pioneering study by Waldmann et al, the impact of anomeric configuration of the glycosyl carbamates on their enzymic hydrolysis process was studied (*Carbohydr. Res.* **1998**, 305, 341-349). Interestingly, they found that both the α - and β -glycosidase actively cleaved the α - and β -anomers of the glycosyl carbamates, respectively, with high stereospecificity (please see Fig. 1D), these results demonstrate that the anomeric purity of the carbamate linkage is an important issue in glycobiology, especially for medical developments. This example has been added in the revised introduction section. For details, please see page 3, paragraphs 2, and revised Fig. 1.

3. The authors have proposed carbamate linkage could offer a potential route to synthesize prodrugs. It would be interesting to discuss the kinetics of enzyme-catalyzed hydrolysis of α - and β -carbamate linkages, given that the authors have primarily constructed β -linkages.

Answer: The kinetics of enzyme-catalyzed hydrolysis of α - and β -carbamate linkages has been studied and discussed in a pioneering work by Waldmann et al (Carbohydr. Res. 1998, 305, 341-349). This example has been added in the revised introduction section. For details, please see page 10, paragraphs 2 and 3, and revised Fig. 3.

4. One section of this manuscript has been dedicated to constructing glycoconjugates using amines as drug molecules. The authors should show at least few examples where sugar confers cell-specific targeting, controlled release or improve cellular internalization.

Answer: Thanks for this comment and this is an important point. The use of glycoconjugates of amines in drug development has attracted a great deal of research interest, and the role of the sugar moiety in the pharmacological activities of the *N*-glycomodified amine-containing drugs have been studied previously in several cases, demonstrating that *N*-glycomodification significantly enhanced drug targeting and specificity. We have selected one example to add in the introduction section (in Fig. 1C), and briefly introduced this work. More examples are cited in Ref. 23 & 24. For details, please see page 3, paragraphs 1, and revised Fig. 1.

5. Have the authors tried trapping the carbamate ions with other electrophiles such as alkyl halides (Cl, Br, I), carbocations, carbonyl compounds, sulfonium ions, or nitronium ions?

Answer: In recent years, the reaction between amines and CO₂ has attracted considerable attention, and this chemistry has been reviewed recently (please see Ref 43). Previous studies revealed that carbamate anions could be trapped by simple active alkyl halides such EtI. To the best of our knowledge, trapping of carbamate anions by carbocations, carbonyl compounds, sulfonium ions, or nitronium ions have not yet been

realized, and this will be a subject of our further research. Thanks again for this suggestion.

Minor Comments:

1. In lines 59 and 60, the authors mentioned that the carbamate linkage enhanced the water solubility of molecules. Please provide specific references that discuss similar molecules where water solubility has been enhanced after the introduction of a carbamate linkage.

Answer: Thanks for the suggestion. The effect of *N*-glycosyloxycarbonylation modifications on increasing the aqueous solubility of amine containing drugs has been demonstrated in several literatures (Ref. 19). These references have been added.

2. In lines 90 and 91, “halide ions” either can be removed since glucopyranosyl halides is generic terms for all halides or be specific which halides has been used.

Answer: Thanks for the suggestion. This has been fixed.

3. In lines 98-100, the authors describes “It was observed that the anomeric configuration, types of protecting groups, and halide ions in the glucosyl donors considerably influenced the coupling efficiency”, the authors need to discuss the effects of different types of protecting groups and their positions on sugar molecules, particularly how these factors influence the stereochemistry and yield of the products.

Answer: Thanks for the suggestion. A summary schematic diagram of the scope of glycosyl halides has been provided (Supporting information, section 4, page 14), and a detailed discussion on the effect of the types of glycosyl donors and protecting groups on the reaction yields and stereoselectivities has been provided. For details, please see page 5, paragraphs 2, and revised Fig. 2.

4. The authors have drawn a conclusion of S_N2 character of the reaction, most of the halides used in the manuscript are α -anomer. It is generally accepted that β -anomer (halides) is more reactive than α , it would be fascinating to know if β -anomeric halides

are used S_N2 character of the reaction would be retained. There are multiple methods known for preparing β -anomeric halides, such as the bromination of thioglycosides.

Answer: Thanks for the suggestion. The reactivity of the glycosyl halides bearing equatorial anomeric C-X bond was studied using **G31** as a model substrate; poor yield and an anomeric mixture of the products were obtained (Fig. 3B). This result is consistent with those in the literature (Ref 58), where it was found that β -glucosyl chlorides exhibited much poor S_N2 reactivity than its α -counterpart toward carboxylate anion nucleophiles. For details, please see page 7, paragraph 1 and revised Fig. 3.

5. The conclusion section is poorly written, with significant portions being repeated verbatim from the abstract, such as the statement, “The mild reaction conditions and high chemoselectivity have allowed successful N-glycoconjugation of 100 amines, including 36 drug molecules.”

Answer: Thanks for the suggestion. The conclusion and abstract section have been revised to avoid simple repetition.

Including all the comments and suggestions would have a broad impact considering the general readership of Nature Communications journal. The potential publication of this manuscript in Nature Communications is only recommended if all the comments and suggestions are addressed.

Thank you again for your careful review of our manuscript and for your constructive questions and suggestions. We have made improvements to the manuscript as per your request, and hope that we have answer the questions and fulfil the requirements you have raised.

Reviewer #4:

In this manuscript, the authors present a new method for N-glycoconjugation of amine-containing drugs via a carbamate linkage. The process involves pre-treating the amines under a CO_2 atmosphere in the presence of bases, followed by reaction with glycosyl halides in the same flask at room temperature. Demonstrated with over 100 examples,

the reaction is highly anomerically selective and straightforward, representing a significant improvement over existing methods for preparing similar carbamate-linked N-glycoconjugates.

Additionally, the authors show that the corresponding glycoside-conjugate of Crizotinib can be hydrolyzed in vivo to yield the parent Crizotinib, with comparable antitumor activity to the parent drug. Given that some sugars are used as directing groups for drug delivery, this reviewer is curious whether a carefully selected sugar-conjugate could also influence the distribution of the conjugated drug. Moreover, could 1,2-anhydro sugars be used instead of glycosyl halides in this transformation?

Overall, this biotransformation-inspired glycoconjugation is highly anomerically selective, the reaction operation is not technically demanding, and the resulting glycoconjugates are enzymatically cleavable, potentially useful in medicinal chemistry. The manuscript is recommended for publication in Nature Communications after addressing the following questions.

We are deeply grateful for the comments provided by the reviewer regarding our manuscript. In the following section, we provide a point-by-point response to each of the comments. We hope that our responses adequately address your concerns.

a) Given that some sugars are used as directing groups for drug delivery, this reviewer is curious whether a carefully selected sugar-conjugate could also influence the distribution of the conjugated drug.

Answer: The role of the sugar moiety in the pharmacological activity of the *N*-glycomodified amine-containing drugs have been studied previously, demonstrating that *N*-glycomodification significantly enhanced drug targeting and specificity. A representative example (Ref. 30) is shown in Fig. 1, where the Hecht group confirmed that the disaccharide moiety of bleomycines (BLM) facilitated its uptake by cancer cells, thus BLM carbamate glycoside (**5**) exhibited significantly more cytotoxic activity than deglycoBLM (**6**). For additional examples, please see Ref. 23 and 24.

b) Moreover, could 1,2-anhydro sugars be used instead of glycosyl halides in this transformation?

Answer: Thanks for raising this question. In order to answer this question, we prepared glycosyl epoxide **G30** and tested its reactivity toward carbamate anions, and unfortunately no desired *N*-glycoconjugated product could be detected. This set of data has been added to the revised manuscript (Fig 2, entry 40). For details, please see page 6, paragraph 1.

1. Since the use of bases in the system, is there any analysis regarding the potential racemization of optically active compounds, such as α -amino acids or similar moieties in drugs?

Answer: Thanks for raising this important question. We have carefully analyzed the NMR data of all the *N*-glycoconjugated products synthesized from α -amino acids or similar moieties in drugs, and no racemization of the products was detected.

2. Do the authors have any speculation about which type of enzyme catalyzes the hydrolysis of the *N*-glycoconjugates?

Answer: Thanks for raising this question. The enzyme-catalyzed hydrolysis of carbamate glycoside linkages has been studied in detail. It was revealed that the anomeric configuration of 1-*O*-glycosyl carbamates significantly influenced the enzymatic cleavage process, both the α - and β -glycosidase actively cleaved the α - and β -anomers of glycosyl carbamates respectively with high specificity (Ref. 20). In recent years, it was gradually realized that carbamate glycosides bearing different types of sugar moiety could be recognized and selectively hydrolyzed by different types of glycosidases. Thus, a series of different types of carbamate glycosides as glycosidase probes were developed, including hexosaminidase probe (Ref. 35), poly- β -(1 \rightarrow 6)-*N*-acetylglucosamine (PNAG) glycosidase probe (Ref. 34), and β -galactosidase probes (Ref. 31 and 32).

3. Line 59, it seems not “a carboxyl (-COO-) linkage”.

Answer: Thanks for pointing out this. This has now been corrected as “a *carbonyloxy* (-COO-) linkage”.

4. As declared in line 115-117, “1,2-trans-substituted pyranosyl halides” and several other halides were not suitable for the titled transformation. Since G22 is bearing anomeric iodide, G25 is bearing anomeric bromide, corresponding anomeric chlorides have not been tested, it is better not to draw a general conclusion for 1,2-trans-substituted pyranosyl halides or 2-N-substituted pyranosyl halides.

Answer: Thanks for pointing out this. This section has been revised accordingly, for details, please see page 5, paragraph 2.

5. Line 190, “... crizotinib was quickly at the similar level...” is not a complete sentence.

Answer: Thanks for pointing out this. It has now been corrected as “...*crizotinib was quickly increased to the similar level ...*”

6. The organization of Table S1 need be reordered to better present the optimization strategy.

Answer: Thanks for the suggestion. Table S1 has now been reordered to better present the optimization strategy.

7. Some ¹H NMR spectra may need to be further purified such as GA1, GA2, GA4, GA122; and some NMR spectra have uneven baselines, such as ¹³C NMR of GA4, GA13, GA65, GA69, GA73, GA77, GA80, GA81, 82, GA83, GA103.

Answer: Thanks for pointing this out. **GA1, GA2, GA4, GA122** have been further purified to attain high-quality ¹H NMR spectra. The baselines in ¹³C NMR of **GA4, GA13, GA65, GA69, GA73, GA77, GA80, GA81, GA82, GA83, and GA103** have been revised to attain spectra with flat baselines.

Response to the comments and suggestions

Reviewer #1:

The authors have addressed my concerns adequately. I support its publication now.

Thank you for your positive comments on our revised manuscript. We greatly appreciate your recommendation to publish our manuscript in Nature Communications.

Reviewer #2:

The authors have responded well to the questions I asked. This work provides a new general approach for Chemical N-glycoconjugation modification of drugs. The experimental methodology is novel and rational, and the experimental results support the conclusions.

Thank you for your positive comments on our revised manuscript. We greatly appreciate your recommendation to publish our manuscript in Nature Communications.

Reviewer #3:

The revised manuscript by Professor Yu and colleagues addresses most of the comments and suggestions provided. The authors have underscored the importance of the carbamate linkage by referencing research from the Hecht group, which demonstrates how drug uptake can be enhanced by sugar moieties.

However, I disagree with their rebuttal on the transformation of oligosaccharides into their peracetylated forms for the generation of anomeric halides. In my view, this transformation may require multiple steps depending on the protecting and anomeric groups present on the sugar, and thus, their justification for synthesizing anomeric halides from peracetylated oligosaccharides in a single step seems overly simplistic. Additionally, the deprotection of acetates might compromise the carbamate linkage.

Regarding the enhancement of drug solubility, the increase in solubility is not due to the carbamate linkage but rather the hydroxyl groups on the sugar moiety, which is an inherent characteristic of sugars (as also mentioned in reference 19). The way the authors have presented this in their paper is somewhat confusing, as it suggests that the

improved solubility is a direct result of the carbamate linkage. However, I do not believe this issue warrants further explanation.

I have no additional comments or suggestions, and I recommend accepting the article without further revision.

Thank you for your positive comments on our revised manuscript. We greatly appreciate your recommendation to publish our manuscript in Nature Communications. For some types of oligosaccharides, such as maltotriose, cellulosic oligosaccharides, etc., it is indeed possible to obtain fully acetylated glycosyl bromides by a two-step process of first fully acetylation followed by bromination. but for more types of glycans, which are often difficult to purchase commercially, it does require, as you say, a multi-step synthesis to prepare them.

The enhancement of the water solubility of 1-*O*-Glycosyl carbamates is indeed due to the glycosyl portion rather than the carbamate fragment, and to make this clear we have made some changes, and it now reads: "1-*O*-Glycosyl carbamates, which combine amine and sugar through a carbonyloxy(-COO-) linkage, usually exhibit enhanced water solubility (compared to parent amines)¹⁹ due to the presence of sugar moiety."